# Optimization of the Physical, Optical and Mechanical Properties of Composite Edible Films of Gelatin, Whey Protein and Chitosan

**DOI:** 10.3390/molecules27030869

**Published:** 2022-01-27

**Authors:** Selene Elizabeth Herrera-Vázquez, Octavio Dublán-García, Daniel Arizmendi-Cotero, Leobardo Manuel Gómez-Oliván, Hariz Islas-Flores, María Dolores Hernández-Navarro, Ninfa Ramírez-Durán

**Affiliations:** 1Laboratorio de Alimentos y Toxicología Ambiental, Facultad de Química, Universidad Autónoma del Estado de México, Paseo Colón Intersección Paseo Tollocan s/n. Col. Residencial Colón, Toluca 50120, Estado de México, Mexico; herrera.v.selene@gmail.com (S.E.H.-V.); lgolivan74@gmail.com (L.M.G.-O.); harizislas@uaemex.mx (H.I.-F.); lolyhn@yahoo.com.mx (M.D.H.-N.); 2Departamento de Ingeniería Industrial, Facultad de Ingeniería, Campus Toluca, Universidad Tecnológica de México (UNITEC), Toluca 50160, Estado de México, Mexico; arcoda21@gmail.com; 3Laboratorio de Microbiología Medica y Ambiental, Facultad de Medicina, Universidad Autónoma del Estado de México, Paseo Tollocan intersección Jesús Carranza s/n, Toluca 50120, Estado de México, Mexico; ninfard@hotmail.com

**Keywords:** gelatin (G), whey protein (W), chitosan (C), surface response methodology (SRM), composite edible film, food packaging

## Abstract

The aim of this work was to evaluate the effect of the concentration of gelatin (G) (3–6 g), whey protein (W) (2.5–7.5 g) and chitosan (C) (0.5–2.5 g) on the physical, optical and mechanical properties of composite edible films (CEFs) using the response surface methodology (RSM), as well as optimizing the formulation for the packaging of foods. The results of the study were evaluated via first- and second-order multiple regression analysis to obtain the determination coefficient values with a good fit (R ˃ 0.90) for each of the response variables, except for the values of solubility and b*. The individual linear effect of the independent variables (the concentrations of gelatin, whey protein and chitosan) significantly affected (*p* ≤ 0.05) the water vapor permeability (WVP), strength and solubility of the edible films. The WVP of the edible films varied from 0.90 to 1.62 × 10^−11^ g.m/Pa.s.m^2^, the resistance to traction varied from 0.47 MPa to 3.03 MPa and the solubility varied from 51.06% to 87%. The optimized values indicated that the CEF prepared with a quantity of 4 g, 5 g and 3 g of gelatin, whey protein and chitosan, respectively, provided the CEF with a smooth, continuous and transparent surface, with L values that resulted in a light-yellow hue, a lower WVP, a maximum strength (resistance to traction) and a lower solubility. The results revealed that the optimized formulation of the CEF of G–W–C allowed a good validation of the prediction model and could be applied, in an effective manner, to the food packaging industry, which could help in mitigating the environmental issues associated with synthetic packaging materials.

## 1. Introduction

The most widely used materials for packaging food, cosmetics and pharmaceutical products are synthetic plastics [1,2] due to their low production cost, light weight and excellent protection that they provide to the packaged product [3]. However, as they are synthetic compounds, with delayed biodegradation (100–700 years), and oil derivatives, their use poses serious ecological problems, such as the emissions released during their manufacture and incineration, the use of resources and the generation of waste [4]. Therefore, there is an urgent need driven by both the industry and consumers to develop environmentally friendly, renewable, naturally derived polymeric materials involving processes that are profitable and minimize pollution and environmental emissions [5,6]. These sorts of natural polymers include polysaccharides, proteins and lipids that have become alternative eco-friendly material for potential elaboration in various applications, such as packaging, paper coating, biomedical, food science and agriculture [2], as well as edible films [7]. These types of raw materials include various by-products of earlier technological processes related to agriculture and food processing [8]. Among the materials obtained from agroindustrial by-products, which are employed in the elaboration of edible packaging, extruded corn, post-extraction rapeseed meal, dried fruit pomace (chokeberry, currant, apple and raspberry pomace) and microcrystalline cellulose (MCC), whey protein, gelatin, chitosan, soy protein and zein can be found, among others [8,9,10,11,12,13,14]. Gelatin is a protein of animal origin, which is mainly obtained from porcine or bovine by-product. Its helicoidal triple structure is characterized by repeated sequence chains of amino acid triplets (glicine-X-proline, where *X* represents alanine, lysine, arginine, methionine or valine) [15]. This protein has a wide range of applications in the food industry due to its abundance and its functional properties as a thickener, foaming agent, stabilizer, its capacity to form biodegradable films and as a micro-encapsulating agent [10]. Gelatin films have been applied to a variety of foods to extend their shelf life [16,17].

Whey protein is a subproduct of the cheese industry [18]. It is composed of different fractions of protein: β-lactoglobulin (β-Lg), α-lactoalbumin (α-La), serum albumin (BSA) and immunoglobulin (Ig) [19,20]. Due to its characteristic functions, it has been employed in the food industry as an emulsifier [13,14,21,22], dietary supplement and has been studied as a raw material for the preparation of CEFs [23,24]. Protein films are characterized by presenting good mechanical properties, as well as being excellent as barriers to gases [25]; however, they present little resistance to water, tend to present a higher water vapor permeability than other biopolymers and lack antimicrobial properties [26].

An effective strategy to overcome these limitations, by improving the functional properties of the films prepared with these proteins, is to develop CEFs through mixing gelatin, whey protein and other polymers from renewable sources, such as chitosan [27].

Although the costs of chitosan are considered high, it is a non-toxic polysaccharide, which is obtained from the deacetylation of the chitin that is extracted from the exoskeleton of crustaceans, insects and the cell walls of microorganisms [20,28]. It is a co-polymer of N-acetyl-D-glucosamine and D-glucosamine, with the latter being superior to 60% [29,30]. The biodegradability, biocompatibility, excellent film formation capacity [13,14] and its antimicrobial properties [31,32] could explain its use in a wide range of applications that may include areas such as pharmaceuticals, biotechnology, biomaterials, biomedicine, agriculture and food processing [33], so using it as a material for the development of a biodegradable bioplastic is a cost benefit with a great impact on the environment compared to polyethylene-based plastics that take up to 700 years to degrade [4]. The biodegradable plastics market is projected to grow from over 2.0 billion USD in 2015 to 3.4 billion USD by 2020, with a growth rate of 10.8% per year [34,35].

Coatings with two independent components have been studied [36,37], but, until now, the interaction of multicomponents in a film has not been studied. Studies have been carried out between pairs of ingredients, such as gelatin–whey protein [11,36], gelatin–chitosan [7,12,13,37,38,39] and whey protein–chitosan [40,41,42], demonstrating that the combination of different proportions of polysaccharides and proteins offers the possibility of fabricating composite films with improved properties in order to satisfy the expectations of the consumer [36]. Nonetheless, until this moment, there are no records on composite films consisting of gelatin, whey protein and chitosan, and, thus, there is a need to study the relationship between these components since the composition of the film affects its structure and properties [27]. The SRM allows understanding the individual relationship and the interaction between multiple factors based on physical responses [43,44,45].

The aim of this work was to evaluate the influence of the levels of gelatin, whey protein and chitosan over the physical, optical and mechanical properties of CEFs, with the final goal of optimizing the formulation for the packaging of foods via surface response methodology (SRM).

## 2. Results and Discussions

### 2.1. Adjustment of the Surface Response Methodology Model

The influence of the concentrations of gelatin (G), whey protein (W) and chitosan (C) over the physical (T, MC, WAC, S, WVP), optical (L*, a* and b*) and mechanical (YM, ST, E) properties of the CEFs was analyzed through the surface response methodology using the Box–Behnken design. The different formulations of the Box–Behnken design and the response variables are shown in Table 1 and Table 2, respectively.

The analyses of variance (ANOVAs) of the response variables, to determine if the quadratic model is or is not significant, are shown in Table 3.

In Figure 1 are response surface plots showing the interaction effects of the process variables that had the best response. In Figure 2 and Figure 3, the response surfaces are shown, which complement the remaining interactions of each of the response variables.

The suitability of the model was determined via the determination coefficient (R^2^), and the lack-of-fit test was employed to evaluate the adjusted model and to determine if the model was adequate for the prediction of the responses. The viscosity results indicate that the model explains 99.1% of the real values of the film-forming solution. Moreover, the lack-of-fit test was significant up to 5%, which indicates that it is a good model and adjusts well to the experimental data.

The estimated R^2^ values show to be satisfactory for MC (93.9), T (93.9), WAC (93.7), S (88.8), WVP (99.4), TR (92.2), L (95.2), a* (93.1), ST (97.3) and YM (96.5). The high R^2^ values indicate a high degree of correlation between the experimental and predicted response values. The lack-of-fit *p*-values calculated for the parameters were significant (*p* > 0.05; Table 3), which reveals that the mathematical model was successful in the prediction of said variables, except for %S and %WAC, with lack-of-fit *p*-values of 0.002 and 0.048, respectively, which means that the model, adjusted in such a manner, does not adequately represent that data.

### 2.2. Determination of the Second-Order Polynomial Mathematical Models

The experimental data of the parameters were analyzed via multiple regression to obtain the second-order polynomial mathematical models. These mathematical models allowed for the demonstration of the relationship between the independent variables with the response variables of the CEF of G–W–C. Coefficients of regression for the models adjusting the physical, optical and mechanical variables of the CEFs are presented in Table 3.

### 2.3. Effect of the Independent Variables over the Viscosity of the Edible Film-Forming Solution Composed of G–W–C

The viscosities of the film-forming solutions were found within an interval of 26.53 to 542.53 mPa.s. The mathematical model showed that the linear, interaction and quadratic effects were significant regarding the viscosity results, except for the quadratic effect of gelatin and the W–C interaction, which did not show a significant effect (*p* < 0.05). Factors G, W and C and interactions G–W, G–C, G–G and W–W showed a positive effect over viscosity (Figure 1A). Where the C–W interaction is observed, it has been reported [15] that the concentration of protein in film-forming solutions affects viscosity due to the interactions between the polymers, while the addition of chitosan to the solution interferes with the formation of the protein network because of the G–C interactions. The enhanced mechanical properties of protein–polysaccharide blends are attributed to the multiple and strong intermolecular interactions (by hydrogen bonding, dipole–dipole link formation and charge effects) between the hydroxyl groups of the polymer chains. In addition, cross-linking with thermal treatment allows the generation of bonds between the chains of proteins and polysaccharides (products of Maillard reactions), which improves the mechanical properties of the polymer network [34,37]. Hydrophobic forces could play a major role in the binding between C and W [46,47]. At pH values lower than 6.3, the amino groups are protonated (NH^3+^), which allows C to make electrostatic contacts with anionic proteins, binding to the OH^−^ groups of the proteins [48], which leads to an increase in the viscosity of the composite solution.

### 2.4. Effect of the Independent Variables over the Physical Properties of the Composite Edible Films of G–W–C

Homogenous, thin, and flexible edible films were obtained from the 17 formulations of the Box–Behnken design, with a thickness ranging from 0.26 to 0.39 mm. The thickness indicates the quality of the CEF and is related to the physical, barrier and mechanical properties of the dry CEF [7]. The mathematical model for thickness shows that the nine effects are significant to a 95% confidence interval. The results indicate that the thickness increases with the content of gelatin, whey protein, chitosan and the interaction of the components; however, the quadratic effects of gelatin and whey protein indicate a maximum of this component for the acquisition of a greater thickness (Figure 1B). The moisture content (MC) present in the films indicates the hydrophilia of the films, with the most hydrophilic films being those that present the highest values of moisture content [31]. The data show that gelatin, chitosan and the quadratic effect of whey protein and chitosan had a significant negative effect (*p* < 0.05) over MC, while the interactions between the independent variables (G–W, G–C and W–C) presented a positive effect. Figure 1C represents the effect of the variation of the concentrations of whey and chitosan on the WVP of the films. The whey protein and the quadratic effect of gelatin did not present a significant effect over MC. The moisture content changed from 30.04% to 17.92% with the content of gelatin and chitosan in the film (Figure 2D). The increase in gelatin and chitosan content in the packaging increased the reticulation between both biopolymers, leading to a greater molecular cross-linking that contributed to a lower retention of the water molecules during the drying of the films. According to another study [12], the G–C interactions are produced by electrostatic and hydrogen bond interactions, which occur thoroughly between the -COOH, -NH_2_ and -OH groups of the amino acids in gelatin and the -OH and -NH_2_ groups in chitosan. Elsewhere [49], the properties of chitosan-based films and composite films were studied based on chitosan–gelatin. The authors found that the gelatin–chitosan composite film presented a low MC, while the highest value was observed for the chitosan-based films, which is also the case in the present work. The water absorption capacity (WAC) of the films is an essential aspect in the application they are designed for, such as when they are utilized as packaging materials for foods rich in water (e.g., fish fillets, meat, third- and fourth-tier sliced fruit, etc.). The analysis of the WAC tests of the edible films was influenced by three independent variables (*p* < 0.05; Table 3). It was demonstrated that the interactions between the independent variables G–W were significant, while those of the independent variables (G–W and W–C) were not significant (*p* > 0.05; Table 3). The values of this parameter were positively influenced by the quadratic effect of G, W and C. The linear factors of W and C decreased the WAC of the film from 132.36% to 51.38% (Figure 2E,F). The increase in the components increased the interaction of the network between the biopolymers and simultaneously decreased the interaction of the biopolymers with the water to form hydrogen bonds. The incorporation of chitosan granted a greater hydrophobicity to the films. Chitosan exhibits poor solubility in neutral and basic media, limiting its use in such conditions, but is soluble in aqueous acidic media via primary amine protonation [50]. The presence of large amounts of protonated -NH_2_ groups on the chitosan structure accounts for its solubility in acid aqueous media since its pKa value is approximately 6.5 [51], while the whey protein had a higher affinity for water absorption due to its hydrophilic nature, having its hydroxyl groups exposed and available for interaction with water [21] or other compounds as impurities.

Solubility is an indirect indicator of the biodegradability of the films. Its ideal value depends on the application one wishes to give it. A high water solubility is desirable for films when their application is for the packaging of foods of immediate consumption or to encapsulate additives, while a low solubility is adequate to package foods with a high water activity.

The solubility of the CEF increased from 24.25% to 45.87% with the addition of whey protein and decreased with the content of gelatin and chitosan (Figure 1E and Figure 2G,H). The linear and quadratic coefficients of all the variables were significant (Table 3) for solubility, except for the gelatin–chitosan interaction.

The whey protein–chitosan interaction promoted a low solubility, while the gelatin–whey protein interaction led to an increase (Table 2). The solubility of the film is most likely due to the high affinity of gelatin for water, which would signify a greater solubility, with its hydrophilic groups being exposed; however, the increase in the concentration of chitosan promoted the formation of a compact and strong network. This could be due to different behaviors, such as the formation of an interpenetrating polymer network, which is a combination of two crosslinked polymers [52]. Films consisting of two or more polymer networks interpenetrate, providing a first brittle network, comprising densely crosslinked polyelectrolyte chains, while the second is ductile and comprises weakly crosslinked nonionic polymer chains, which allows outstanding mechanical properties to be obtained that are much better than the sum of the mechanical properties of its network components separately [53]. For example, in previous studies carried out in an experiment with mixtures, it was observed that 100% chitosan films were weaker than those with the mixture of 50% protein and 50% chitosan. The film strength could also be due to the interaction of the hydrogen bonds between the whey protein and chitosan, which produced a greater physical interference between the polypeptide bonds of gelatin and the water molecules within the film matrix. Said interference could be the main effect of the reduction of water solubility in the composite films [5]. The effect of chitosan over the films is according to a study [54] in which it was observed that the increase in chitosan content within the eggshell membrane gelatin film decreased the solubility from 56.28% to 48.33% due to a certain level of electrostatic forces and interaction of hydrogen bonds between the chitosan and gelatin. The results were also consistent with those reported elsewhere in the literature [12,49]. Nonetheless, the quadratic effect of chitosan had a positive influence over solubility, while that of gelatin and whey protein presented a negative influence; in other words, an excess in the composition materials of the film could lead to the formation of chitosan networks and allow the interaction of gelatin and whey protein with water, promoting the dissolution of the materials of the protein-based film. All the linear and interaction effects indicated a significant positive influence for the PVA of the films, with values ranging from 1.86 × 10^−^^8^ g/msPa to 7.39 × 10^−^^8^ g/msPa, due to the hydrophilic nature of the hydrocolloids that make up the film.

### 2.5. Effect of the Independent Variables over the Optical Properties of the Edible Films Composed of G–W–C

The optical properties of the films are important in the application they are designed for since they can alter the appearance of the food and directly influence the acceptance of the product by the consumer, especially if the film is used as a coating [7]. The colors of the films were evaluated via the Hunter system, where L* represents the luminosity, a*− represents green and b*+ represents yellow. The edible films prepared with different concentrations of gelatin, whey protein and chitosan resulted in visually homogeneous, translucid and bright films. Table 2 shows the data obtained from the design for the Hunter parameters (L*, a* and b*), while Table 3 contains the linear effect coefficients of each one of the independent variables, its interactions and the quadratic effect.

The linear, interaction and quadratic effects were significant regarding the results of luminosity (L*). The factors W, C, G–C, G–G and W–W presented a positive effect over luminosity, while G, G–W, W–C and C–C exhibited an antagonistic effect (Figure 1G and Figure 3A,B). The range of the L* values of the films obtained was from 8.02 to 15.74.

As can be seen in Table 2, negative a* values were obtained (from −2.62 to −3.57), which suggests that the films had a light-green hue. The analysis of variance showed that the linear coefficients of C, the G–C interaction and the quadratic factor of G–G and W–W presented a positive effect, while coefficients of G and W, the G–W interaction and the quadratic factor of C–C showed a negative effect except for the W–C interaction, which did not show a significant effect (Figure 1H and Figure 3C,D).

The b* parameter adopted positive values (ranging from 18.12 to 34.23), which suggests that the films present a light-yellow hue. Table 3 shows that the value of b* was significantly influenced by all the linear coefficients (Figure 1I and Figure 3E,F); however, the linear term of gelatin had an antagonistic effect. It was possible to observe that b* showed the highest values for the lowest concentrations of chitosan, exhibiting an opposite tendency to that observed by the L response, although this value did not experience a change regardless of the different proportions of each film-forming polymer. According to another paper [49], the yellow coloration of chitosan in pure films is due to the presence of repeated units of 2-amino-2-deoxy-D-glucopyranose joined together by 1,4 beta bonds, also observed by others [7,55].

The behavior of the L* parameter was like that observed in another study [39], with a value of 0.91 for films based on C and G from pigskin, while the a* and b* parameters presented the contrary effect since it was demonstrated that the chitosan decreased the values of a* and increased the values of b*, as was the case with chitosan- and whey protein-based films [40]. This could be due to the gelatin–whey protein interaction reported in this work, which is non-existent in prior research.

The transparency of any material is indicative of the degree of light that goes through it and is related to the speed of oxidation of the lipids. The different formulations based on chitosan, gelatin and whey protein produced films with different degrees of transparency (Table 2). The addition of whey protein and chitosan had a significant negative effect on the values of transparency of the films; in other words, it increased the transparency of the films (Figure 3H). However, the gelatin did not present a significant linear effect. Moreover, the various interactions of the three components (G–C and W–C) did not have a significant influence on this parameter. Nonetheless, the transparency values were positively influenced by the quadratic effect of the gelatin and whey protein. The transparency values of the films decreased from 8.14 to 3.90 upon an increase in the concentration of gelatin, most likely due to the greater compaction of the polymeric chain, which modifies the refraction index and restricts the passage of light through the film matrix. The highest values indicate less transparency and greater opacity. The tendency of the transparency values was also reported by other authors (such as [5,37]), who reported transparency values from 0.56 to 0.95 and 0.62 to 1.13, respectively, with films of different chitosan and gelatin composition, and observed that the addition of chitosan increased the transparency of the films. The values obtained were lower than those reported elsewhere [56] for starch-, citric pectin- and Brazilian pine-based films. The films showed values near to the low-density polyethylene films (3.05 at 600/mm) and polypropylene films (1.67 at 600/mm) [57], being a potential packaging material for foods that must not suffer alterations in their appearance.

The crystallinity of the samples could result from the interaction between chains through intermolecular hydrogen bonds and van der Waals bonds that form between the surface of a compound and C (cationic behavior) since it is a mechanical type of bond, which occurs due to the charges in both materials (electrostatic adhesion). In another study [58], this behavior was observed with hydroxyapatite and C.

### 2.6. Effect of the Independent Variables over the Mechanical Properties of the Edible Films Composed of G–W–C

The edible films can maintain and/or improve the mechanical properties of the food for a longer period besides decreasing the physical damage caused during manipulation, transport or storage. For this purpose, so that the film can be employed as packaging, it is necessary to determine the external mechanical resistance and maximum extensibility that support the film and maintain its integrity [59].

The strength (ST) values of the CEFs vary from 15.47N to 86.59 N (Table 2). The analysis of variance of the strength tests of the films revealed that this parameter was negatively influenced by the independent variables of gelatin and chitosan (*p* < 0.05; Table 3). The interaction between the G–W (Figure 1K), G–C (Figure 3I) and W–C (Figure 3J) variables and the quadratic effect of the gelatin and whey protein presented significant positive influence, while the effect liner of W and the quadratic effect of gelatin demonstrated that there were no significant consequences in the ST tests. (Rao et al., 2010) [59] proposed that ST increases with the incorporation of gelatin into the chitosan films. (Mohammadi et al., 2018) [54] found that incorporation of chitosan into the egg membrane gelatin films resulted in stronger films, while the presence of whey protein mixed with chitosan obtained weaker films [40]. The results were similar to those reported by another study [39] in which whey protein was added to the chitosan films.

The gelatin and chitosan improved the Young’s modulus of the films from 0.06 to 0.98. Both independent variables showed a positive effect over Young’s modulus, which means that increasing the content of chitosan and gelatin decreased the elasticity of the polymeric network, making the film more resistant to traction (Figure 3L). (Hosseini et al., 2013) [5] showed an opposite effect, that the addition of chitosan to the gelatin films produced more flexible films, which suggests that chitosan participates in the debilitation or reduction in the number of hydrogen bonds, acting as a plasticizer.

### 2.7. Optimization and Validation of the Box–Behnken Design

With the aim of finding the optimal formulation of the film, to achieve the desired value of the response variables, the simultaneous optimization of multiple responses function was employed. The optimization was carried out on the basis of the following objectives: (1) minimize the moisture content, (2) minimize solubility, (3) maximize transparency and (4) minimize Young’s modulus. It is intended to be used for high-moisture foods; however, this optimization can fit the desired goal so long as the prediction is within the experimentally observed parameters. The optimal levels of the different parameters via the application of the methodology of the desired function in terms of the real values of the raw materials, 6 g of gelatin, 2.5 g of whey protein and 1.48 g of chitosan, were obtained, with an estimated desired value of 0.9258. Subsequently, the estimated parameters were validated with new formulations under the conditions suggested above.

The validation of the model was carried out via the comparison of the predicted results with the experimental results of the edible film composed of C-G–W in optimized conditions (Table 4). The absolute residual error of the dependent variables was found to be between 0.68% and 21.54%, which indicates the validity of the responses obtained through the Box–Behnken experimental design and which are found to be in conformity with the precision of models generated for other films [60,61,62]. The correlation value between the predicted values and those obtained experimentally was 0.9982, which adequately validates the optimization model.

Similar results were found in another study [63] using glycerol, chitosan and drying temperature. A desirability of 1.0 was found; the mean optimized values were for thickness, moisture, solubility, L*, a*, b*, penetrability, density, transmittance and WVTR. (Saberi et al., 2016) [60], using Box–Behnken response surface design, managed to maximize transparency and minimize solubility, yellowness index and moisture content in a pea starch/guar gum edible film, with an overall desirability of 0.77. These characteristics can be effectively produced and successfully applied in the food packaging industry. On the other hand, (Sharma et al., 2016) [43] optimized a film formation following this consideration: (1) minimization of WVP; (2) maximization of tensile strength and (3) minimization of solubility in a sesame protein edible film, showing a good mechanical property, with the ability to be used for the packaging or coating applications of fruits and vegetables, providing a partial barrier to moisture and being considerably effective in reducing moisture loss. (Hajji et al., 2018) [44] observed that edible coatings based on chitosan, glycerol and carotenoproteins, the mechanical and biological properties of composite films using RSM, could be an attractive natural alternative against fungi that attack fruits and vegetables, thereby preventing the occurrence of health and environmental problems. Other studies [64] achieved an optimization of chitosan, starch and glycerol for the physical, mechanical and barrier properties by RSM, showing that the values of the physical and mechanical properties were found to be similar to the predicted values. These results show that this formulation can be applied to prepare the pea starch film with good physical and mechanical properties for further utilization.

As can be seen, the use of RSM for the elaboration of edible coatings could predict future formulations and behaviors to provide alternatives for their application.

## 3. Materials and Methods

### 3.1. Materials

The materials used were bovine Gelatin 290° Bloom (Duche^®^, Toluca, Mexico), whey protein (from the rennet cheese process) (Darigold Inc.; Seattle, WA, USA) and commercial-grade Chitosan (Sigma- Aldrich, Saint Louis, MO, USA) low molecular weight; molecular mass of 50,000–190,000 Da; degree of deacetylation >75%. All the chemical compounds and solvents used were of analytical grade: Glycerol (J.T. Baker, Cd. Mexico, Mexico) and 85% lactic acid (85–90% lactic acid), ACS chloride (Cl) ≤ 0.001 %, trace impurities ACS heavy metals (as Pb) ≤ 5 ppm, ACS sulfate (SO₄) ≤ 0.002 %, ACS chloride (Cl) ≤ 0.001 %. The reagent generally available is a mixture of lactic acid (CH₃CHOHCOOH) and lactic acid lactate (C₆H₁₀O₅) (J.T. Baker, Cd. Mexico, Mexico), which was employed to improve the mechanical properties of the film and increase the solubility of the polymer powders.

### 3.2. Preparation of the Film-Forming Solution

In 100 mL of distilled water, at 70 °C with constant magnetic stirring, chitosan was added and was taken to a pH of 3.5 with lactic acid. Work was done in all the tests to avoid pH effects and complex coacervation and precipitation in case of an opposite charge of two polymers [40], and to improve the solubility of chitosan. Afterwards, gelatin, whey protein and glycerol were added, one by one, allowing the proper incorporation of each of the components (approximately 5–7 min between each one). Temperature and pH were always kept constant. The solution was cooled until it reached a temperature of 40 °C, under constant stirring, and was identified as film-forming solution (FFS). The concentrations of each of the components are described in Table 1.

### 3.3. Viscosity of the Film-Forming Solutions

A method approved by (ASTM, 2005) [65] was employed, using a Brookfield Viscometer (Model RVDV-I, Brookfield Engineering Labs Inc., Mid-dleboro, MA, USA) with a No. 2 needle (spin); the spin was introduced at 60 rpm for 30 s into a 90-mL sample of FFS at a temperature of 25 °C. Afterwards, a measurement was made in centipoise (mPa.s). The viscosity was measured in triplicate for each FFS formulation.

### 3.4. Film Preparation

The composite edible films (CEFs) were obtained using the evaporation casting technique. Five milliliters of FFS were placed in Petri dishes with a diameter of 5 cm. Afterwards, the dishes were kept in a chamber (ITH-75: Lumistell, Gto, México) at 30 °C for 48 h. Prior to analysis, the CEFs obtained were conditioned in the same chamber, with a relative humidity of 50% at 25 °C ± 1 °C for 24 h, as is mentioned in (ASTM, 2004) [66].

### 3.5. Thickness

The thickness (T) of the CEF was measured with a digital micrometer (RS Pro, Fujian, China). The measurements were carried out in six different locations of the CEF. The average value of thickness was used, expressed in millimeters (mm), to perform the WVP and mechanical properties calculations.

### 3.6. Moisture Content

The moisture content (MC) of the CEF was determined according to (ASTM, 2007) [67]. Briefly, a gram of sample was dried for 24 h at 105 °C (until the equilibrium weight was reached). The MC was calculated by weight difference and was expressed as a percentage (%MC). Three replicas were obtained for each sample.

### 3.7. Water Absorption Capacity (WAC)

The water absorption capacity (WAC) of the CEF was determined as described in (ASTM, 2018) [68], with modifications. Briefly, the CEF samples were previously conditioned for 24 h at 50 °C ± 2 °C. Afterwards, they were weighed and submerged in water for 24 h at 20 °C. Then, they were removed from the liquid, the excess surface water was cleaned off the CEF with absorbent paper and these were weighed again. The WAC was calculated by difference of weight and the result was expressed as a percentage (%WAC). Three replicas were tested for each sample.

### 3.8. Water Solubility

The methodology proposed by (Gómez-Estaca, 2011) [12] was employed, with some modifications, to determine the water solubility (WS) of the CEF. All film samples, with a diameter of 5 cm, were dried in a furnace at 105 °C for 24 h [7,69], with the finality of obtaining a constant weight. Afterwards, each sample was immersed in 30 mL of distilled water for 24 h at 20 °C. Then, the water was removed from the CEF via decantation, and these were again dried at 105 °C for 24 h. The water solubility was calculated by weight difference and was expressed as a percentage (%WS).

### 3.9. Water Vapor Permeability (WVP)

The water vapor permeability (WVP) of the CEF was determined via the gravimetric method described in (ASTM, 2013) [70], with some modifications. The different CEFs, previously equilibrated and 5 cm in diameter, were adjoined (with parafilm) to glass containers with dry silica gel (0% RH) and were placed in a Thermo desiccator with distilled water at 30 °C (99% ± 1% RH; 4244.9 Pa at 30 °C). The containers were weighed at time intervals of 1 h for a period of 10 h. The WVP of the edible films was calculated using Equation (1):(1)$$\mathrm{WVP}\left(gm^{-1}s^{-1}Pa^{-1}\right)=wlA^{-1}t^{-1}\Delta P^{-1}$$
where *w* is the increase in weight (*g*) of the container; l is the average thickness of the edible film (*m*); A is the exposed area of the film (m^2^); *t* is the time elapsed (*s*); Δ*P* is the difference in water vapor pressure on both sides of the film (4244.9 Pa at 30 °C). The determinations were carried out in quintuple for each edible film.

### 3.10. Color

The color of the edible film surface was determined via a digital micrometer (RS Pro, Fujian, China). The measurement was carried out by placing the samples over the standard white plate, with number 126,633,047, and the color values were recorded in Hunter scale (L*, a*, b*), where L (luminosity), with values of (−) L representing black and (+) L representing white; (+) a representing red and (−) a representing green; (+) b representing yellow and (−) b representing blue. Five measurements were carried out for each film formulation.

### 3.11. Light Transmission and Transparency of the Films

The ultraviolet (UV) and visible-light barrier properties of the CEF were measured in transmittance mode, with a wavelength from 200 to 800 nm, using a UV-Vis spectrophotometer (UV-5500PC: M&A Instruments Inc., Arcadia, CA, USA) according to the method described by (Shiku et al., 2004) [71]. The measurements were carried out using air as a reference. The transparency values of the edible films were calculated via the following equation [72]:(2)$$\mathrm{Transparency}\;(\frac{A}{\mathrm{mm}})=\frac{-logT_{600}}{x}$$
where *T*_600_ is the fractional transmittance at 600 nm and *x* is the thickness of the films (mm). The high transparency values represent a low transparency of the CEF.

### 3.12. Mechanical Properties

The mechanical properties (strength (ST) and Young’s modulus (MY)) were evaluated via the puncture test, according to the methodology described by (García-Argueta et al., 2013) [11], with some modifications (Figure 4). Briefly, a texturometer (TA.XT Plus Texture Analyser: Stable Micro Systems, Godalming, UK), with a p.0.255 ¼” stainless-steel spherical test probe, a test speed of 1.0 mm/s and a penetration distance of 30.0 mm, was employed to obtain the force–deformation curve (Figure 5). From these curves, the strength and Young’s modulus were calculated. The strength was considered the maximum force prior to the fracturing of the CEF. Young’s modulus (YM) was considered the slope of the linear proportion (6.25 mm up to 50% of the penetration distance of the probe) of the force–deformation curves. The measurements were repeated seven times for each composite edible film. All parameters were determined through the Exponent 6.1.20.0 (Stable Micro Systems, UK).

### 3.13. Experimental Design

The Box–Behnken surface response methodology was used for three independent factors (gelatin, whey protein and chitosan), with five replicas of the central point. Each factor was tested at three levels: low (−1), center (0) and high (1) (Table 1). The range of each component was chosen based on previous studies [11,74] and preliminary trials. The data of the experimental responses of the 17 tests were adjusted to a second-order polynomial model to obtain the regression models:(3)$$Y=\beta_{0}+\beta_{1}C+\beta_{2}G+\beta_{3}W+\beta_{4}C^{2}+\beta_{5}G^{2}+\beta_{6}W^{2}+\beta_{7}CG+\beta_{8}CW+\beta_{9}GW+\varepsilon$$
where the values of *G*, *W* and *C* are the factors of the study (gelatin, whey protein and chitosan), *β*_0_ to *β*_9_ are the regression coefficients for the intersection, simple effects, intersections and quadratic effects, respectively, and ε is the residual.

### 3.14. Statistical Analysis

The experimental results were analyzed via the Statistica 7 software (2004). Multiple regression analysis was used to evaluate the statistical significance of the coefficients of the regression model (*p* < 0.05). The same software was used to generate surface response plots, and the multiple optimizations of response of the process parameters was also carried out. The models were used to study the effects of the various parameters on the response variables (MC, WAC, WVP, TR and YM). The results are expressed as an arithmetic mean.

## 4. Conclusions

The design of the three-level Box–Behnken surface response methodology allowed the successful optimization of the optimal formulation of edible films composed of G–W–C. The concentration of three biopolymers significantly affected the viscosity of the film-forming solution and the physical, optical and mechanical properties of the films. It was found that the optimal formulation of the films contained 6 g of gelatin, 2.5 g of whey protein and 1.48 g of chitosan since they presented a lower moisture content and solubility, greater transparency and better mechanical properties than the rest of the formulations, and these are suggested properties for high moisture foods. The present study suggests that films composed of G–W–C exhibit satisfactory properties for their use as biodegradable packaging materials. These films, being made from biodegradable, non-toxic, edible natural sources, can replace synthetic polymers in certain circumstances, which provides the opportunity to reduce the use of non-degradable synthetic materials in addition to providing options for the development of new plastic products for the pharmaceutical, food and cosmetic industries. The use of RSM for the elaboration of edible coatings could predict future formulations and behaviors to provide alternatives for their application. Nonetheless, additional studies are required to determine the use of the films in commercial food systems.

## Figures and Tables

**Figure 1 molecules-27-00869-f001:**
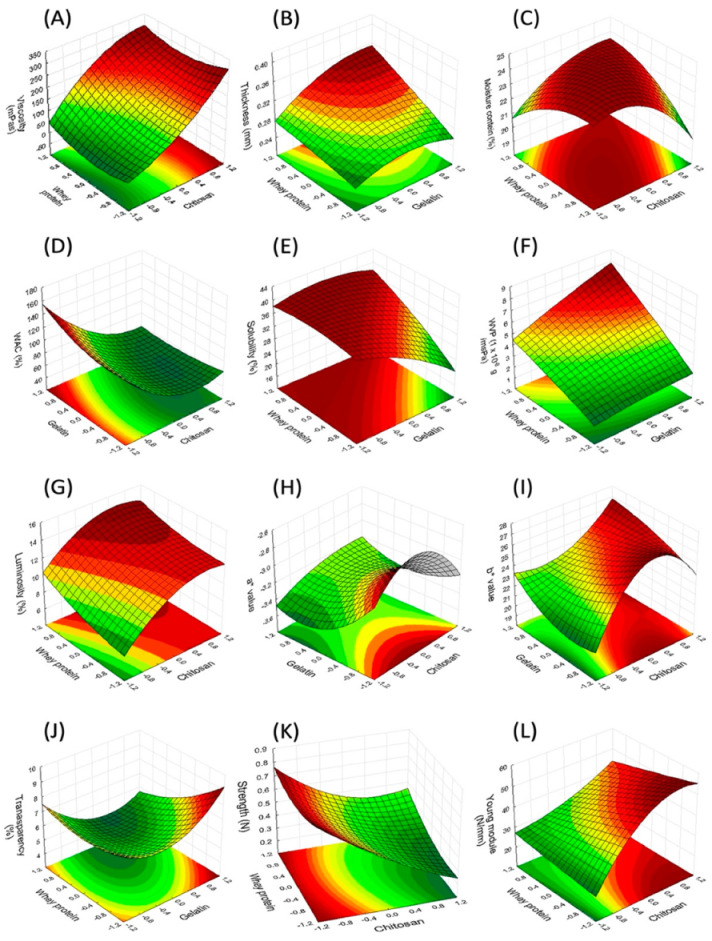
Response surface plots showing the interaction effects of process variables on V: viscosity (**A**), T: thickness (**B**), MC: moisture content (**C**), WAC: water absorption capacity (**D**), S: solubility (**E**), WVP: water vapor permeability (**F**), L: luminosity (**G**), a* value (**H**), b* value (**I**), TR: transparency (**J**), ST: strength (**K**) and YM: Young’s modulus (**L**). −1.0 minimum level of component; 1.0, maximum level of component; 0, mid-level of component. For each surface plot, the level of the third factor is held at its central value. The regions with the dark red color represent the dominant working conditions, ensuring the maximum values for the variables evaluated.

**Figure 2 molecules-27-00869-f002:**
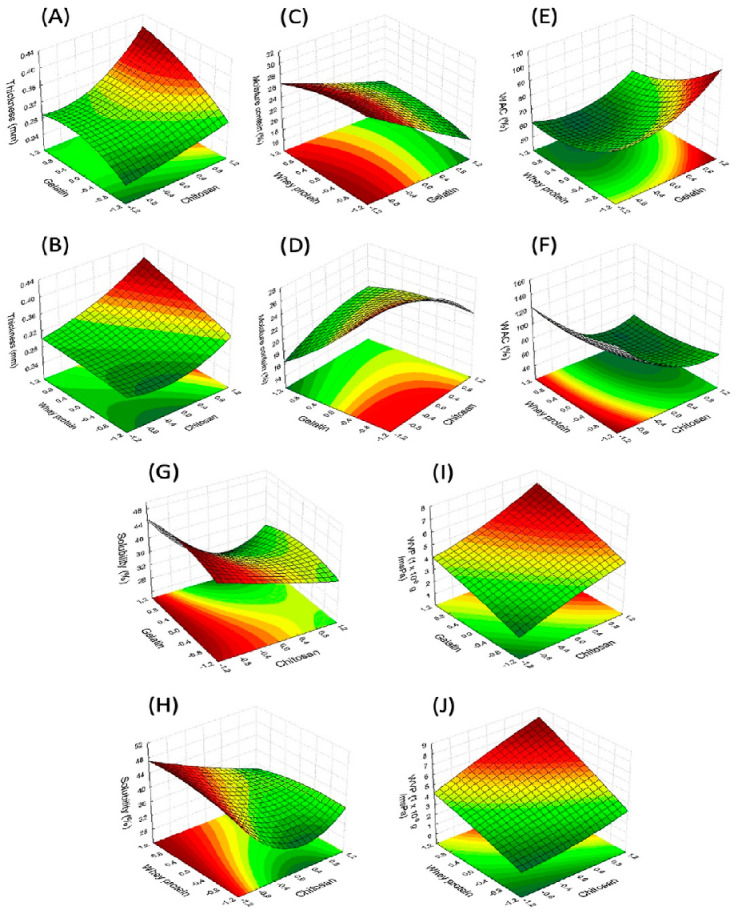
Response surface plots showing the interaction effects of process variables on T: thickness (**A**,**B**), MC: moisture content (**C**,**D**), WAC: water absorption capacity (**E**,**F**), S: solubility (**G**,**H**) and WVP: water vapor permeability (**I**,**J**). −1.0 minimum level of component; 1.0, maximum level of component; 0, mid-level of component. For each surface plot, the level of the third factor is held at its central value. The regions with the dark red color represent the dominant working conditions, ensuring the maximum values for the variables evaluated.

**Figure 3 molecules-27-00869-f003:**
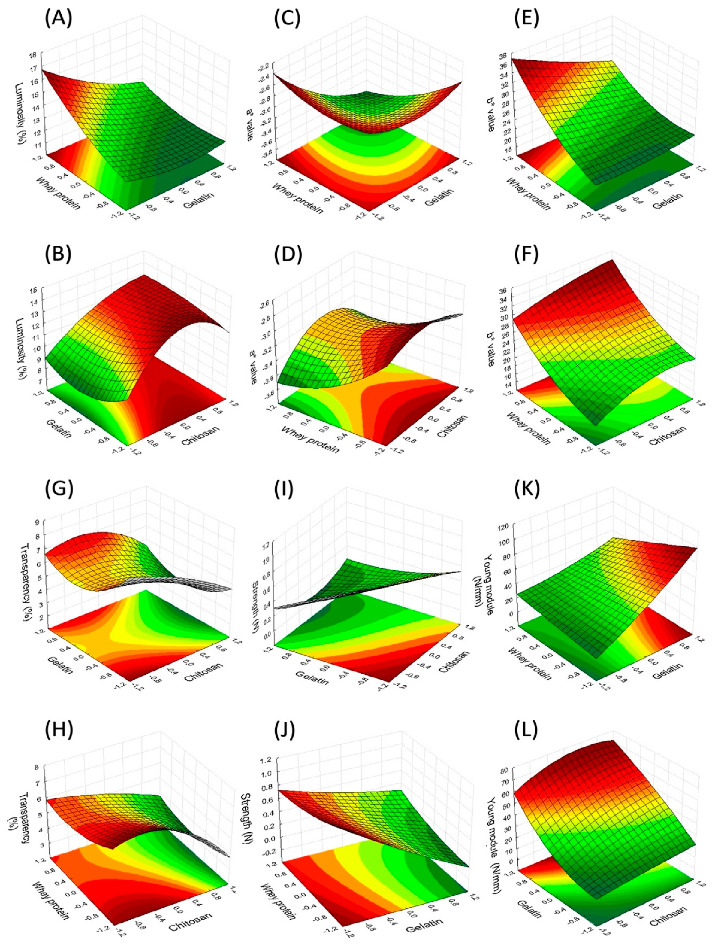
Response surface plots showing the interaction effects of process variables on L: luminosity (**A**,**B**), a* value (**C**,**D**), b* value (**E**,**F**), TR: transparency (**G**,**H**), ST: strength (**I**,**J**) and YM: Young’s modulus (**K**,**L**). −1.0 minimum level of component; 1.0, maximum level of component; 0, mid-level of component. For each surface plot, the level of the third factor is held at its central value. The regions with the dark red color represent the dominant working conditions, ensuring the maximum values for the variables evaluated.

**Figure 4 molecules-27-00869-f004:**
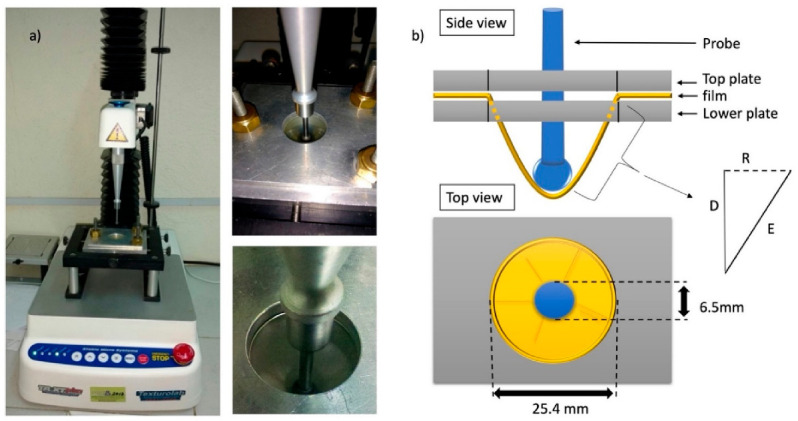
Schematic diagram of the texture analyzer used for compression test for film. The texture analyzer (**a**) consisted of a probe and a support. (**b**) System for measuring the elongation of composite films. D: maximum penetration distance of the probe before rupture (mm). R: radius of circumference of the plate (12.7 mm). E: elongation (mm) of the film [73].

**Figure 5 molecules-27-00869-f005:**
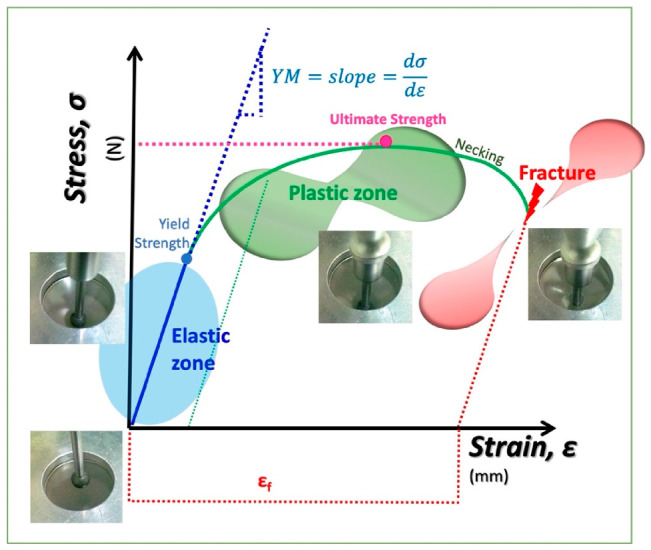
Force–deformation curve [74].

**Table 1 molecules-27-00869-t001:** Coded and uncoded values of independent variables and their levels used for the Box–Behnken design.

	Independent Variables					
Assays	Coded Variables			Uncoded Variables *		
	Gelatin (Factor A)	Whey Protein (Factor B)	Chitosan (Factor C)	Gelatin (g)	Whey Protein (g)	Chitosan (g)
1	−1	1	0	3.0	7.5	1.5
2	−1	0	−1	3.0	5.0	0.5
3	−1	0	1	3.0	5.0	2.5
4	1	0	−1	6.0	5.0	0.5
5	1	0	1	6.0	5.0	2.5
6	0	−1	−1	4.5	2.5	0.5
7	0	−1	1	4.5	2.5	2.5
8	1	1	0	6.0	7.5	1.5
9	0	1	−1	4.5	7.5	0.5
10	0	1	1	4.5	7.5	2.5
11	−1	−1	0	3.0	2.5	1.5
12	1	−1	0	6.0	2.5	1.5
13	0	0	0	3.0	5.0	1.5
14	0	0	0	3.0	5.0	1.5
15	0	0	0	3.0	5.0	1.5
16	0	0	0	3.0	5.0	1.5
17	0	0	0	3.0	5.0	1.5

Factor A, gelatin; factor B, whey protein; factor C, chitosan; −1 = minimum level of component; 1 = maximum level of component; 0 = medium level of component. Glycerol levels were constant in all FFS (6g). * Real values according to design levels of Box–Behnken.

**Table 2 molecules-27-00869-t002:** Box–Behnken experimental design with independent variables (un-coded) and observed values of response variables.

Runs	Responses											
	V (mPas)	T (mm)	MC (%)	WAC (%)	S (%)	WVP × 10^−8^ (g/msPa)	TR	L*	a*	b*	ST (N)	YM
1	96.40	0.29	26.24	58.11	38.95	4.53	6.72	15.74	−2.62	34.23	20.54	0.68
	(0.40)	(0.02)	(0.93)	(8.98)	(2.36)	(0.01)	(0.03)	(0.06)	(0.05)	(0.36)	(0.38)	(0.01)
2	47.33	0.26	30.04	105.06	44.00	2.21	7.53	10.58	−2.98	22.13	22.40	0.98
	(1.22)	(0.01)	(1.76)	(1.08)	(1.85)	(0.00)	(0.04)	(0.18)	(0.02)	(0.62)	(0.30)	(0.09)
3	176.67	0.30	23.68	57.45	35.60	4.32	4.63	12.72	−3.10	25.23	19.76	0.52
	(0.23)	(0.01)	(0.17)	(0.34)	(1.13)	(0.01)	(0.09)	(0.09)	(0.05)	(0.41)	(0.28)	(0.05)
4	210.40	0.29	17.92	132.36	42.15	3.98	6.43	9.26	−3.46	22.63	20.51	0.31
	(0.85)	(0.00)	(0.26)	(19.02)	(0.53)	(0.01)	(0.01)	(0.08)	(0.02)	(0.29)	(0.17)	(0.02)
5	534.33	0.39	21.28	56.25	33.10	6.47	4.66	12.86	−3.24	25.98	18.08	0.18
	(1.30)	(0.01)	(0.74)	(3.79)	(0.83)	(0.02)	(0.05)	(0.50)	(0.07)	(1.18)	(0.14)	(0.00)
6	26.53	0.28	23.90	122.99	41.41	1.86	6.80	8.02	−3.00	18.12	23.11	0.71
	(0.15)	(0.01)	(0.40)	(7.34)	(0.87)	(0.00)	(0.03)	(0.11)	(0.04)	(0.55)	(0.49)	(0.01)
7	266.87	0.32	21.14	63.84	34.90	3.40	4.27	12.41	−3.01	22.34	19.81	0.27
	(1.60)	(0.00)	(0.67)	(1.47)	(0.37)	(0.01)	(0.03)	(0.21)	(0.12)	(0.54)	(0.29)	(0.01)
8	542.53	0.35	20.39	67.12	37.42	7.19	5.45	12.63	−3.55	27.71	18.99	0.27
	(2.50)	(0.01)	(0.86)	(2.73)	(0.32)	(0.02)	(0.03)	(0.12)	(0.01)	(0.36)	(1.12)	(0.02)
9	69.73	0.30	21.63	107.00	45.87	4.33	5.69	10.78	−3.57	27.94	23.29	0.68
	(0.23)	(0.01)	(0.15)	(13.98)	(1.59)	(0.00)	(0.00)	(0.18)	(0.01)	(0.34)	(0.65)	(0.03)
10	285.87	0.39	23.56	51.38	35.57	7.39	3.90	14.13	−3.35	32.21	21.18	0.40
	(0.23)	(0.01)	(0.28)	(2.28)	(2.67)	(0.01)	(0.02)	(0.21)	(0.06)	(1.53)	(0.19)	(0.02)
11	160.67	0.26	28.49	72.69	37.35	2.00	6.90	12.26	−2.69	22.05	21.38	0.90
	(0.83)	(0.00)	(0.41)	(4.74)	(0.43)	(0.01)	(0.01)	(0.47)	(0.06)	(0.86)	(0.38)	(0.01)
12	299.33	0.27	19.04	93.39	24.25	2.68	8.14	11.64	−2.76	22.22	15.19	0.06
	(0.61)	(0.01)	(0.23)	(9.93)	(0.53)	(0.01)	(0.05)	(0.56)	(0.07)	(0.40)	(0.70)	(0.01)
13	157.33	0.30	23.93	61.34	36.55	4.25	5.74	12.55	−3.11	25.54	19.76	0.41
	(0.61)	(0.01)	(0.89)	(5.24)	(0.94)	(0.01)	(0.02)	(0.16)	(0.04)	(0.10)	(0.46)	(0.03)
14	150.80	0.32	23.41	61.42	37.29	4.40	4.77	12.34	−3.33	24.34	21.24	0.41
	(0.40)	(0.00)	(0.20)	(8.12)	(0.71)	(0.01)	(0.03)	(0.40)	(0.04)	(0.85)	(0.27)	(0.02)
15	125.80	0.31	24.32	65.16	36.39	4.29	5.99	12.59	−3.15	25.07	21.53	0.44
	(0.72)	(0.00)	(0.88)	(5.30)	(0.84)	(0.01)	(0.06)	(0.27)	(0.08)	(0.67)	(1.15)	(0.01)
16	150.47	0.31	24.29	59.27	37.93	4.09	5.50	12.11	−3.13	24.05	20.64	0.37
	(8.73)	(0.00)	(1.25)	(5.86)	(0.70)	(0.01)	(0.01)	(0.14)	(0.02)	(0.05)	(1.35)	(0.08)
17	130.40	0.31	24.34	60.59	37.04	4.35	4.74	12.07	−3.15	24.52	20.74	0.39
	(0.40)	(0.02)	(0.59)	(2.29)	(0.46)	(0.01)	(0.41)	(0.26)	(0.07)	(0.40)	(1.00)	(0.06)

Mean ± (SD) n = 3. V: viscosity; T: thickness; MC: moisture content; WAC: water absorption capacity; S: solubility; WVP: water vapor permeability; TR: transparency; L*, a* and b*: CieLab parameters; ST: strength; YM: Young’s modulus.

**Table 3 molecules-27-00869-t003:** Coefficients of regression for the models adjusting the physical, optical and textural variables of composite films.

	Viscosity	Thickness	Moisture Content	Solubility	WAC	WVP (1 × 10^−8^)	Luminosity	a* Value	b* Value	Transparency	Strength	Young’s Modulus
Average Reply (µ)	142.96	0.309	24.057	37.039	61.555	4.274	12.331	−3.173	24.706	5.348	0.406	41.39
A: Gelatin	138.192	0.024	−3.727	−2.372	6.976	0.907	−0.615	−0.203	−0.636	−0.139 *	−0.283	22.08
AA	105.853	−0.016	0.079 *	−1.636	6.374	−0.087 *	0.378	0.152	0.344 *	1.052	0.026 *	3.129
B: Whey (g)	30.142	0.024	−0.091 *	2.488	−8.66	1.689	1.118	−0.204	4.67	−0.545	0.010 *	−5.402
BB	25.92	−0.001	−0.596	−0.912	4.898	−0.087 *	0.359	0.117	1.502	0.404	0.046	0.134 *
C: Chitosan (g)	113.717	0.032	−0.478	−4.282	−29.81	1.151	1.684	0.037	1.868	−1.123	−0.164	8.211
CC	−6.630 *	0.016	−0.904	3.308	19.849	0.060 *	−1.355	−0.175	−1.055	−0.586	0.065	−3.11
AB	76.867	0.012	0.898	2.891	−2.924 *	0.495	−0.622	−0.214	−1.672	−0.629	0.108	−11.059
AC	48.65	0.014	2.429	−0.162 *	−7.126	0.093 *	0.366	0.085	0.064 *	0.282 *	0.083	0.969 *
BC	−6.050 *	0.011	1.174	−0.949	0.883 *	0.381	−0.258	0.058 *	0.016 *	0.184 *	0.041	−2.804
Lack of fit (*p*-value)	0.37	0.545	0.163	0.002	0.048	0.748	0.129	0.892	0.061	0.999	0.624	0.573
Pure Error (MS)	203.3	0	0.399	0.787	26.47	0.023	0.114	0.009	0.941	0.346	0.002	15.25
R^2^	0.991	0.939	0.939	0.888	0.937	0.994	0.952	0.931	0.581	0.922	0.973	0.965
CV	10%	3%	3%	2%	8%	4%	3%	3%	4%	11%	11%	9%

A Significantly different at *p* < 0.05; P: parameter; Model: intercept; A, B and C: linear regression coefficients for gelatin (G), whey protein (W) and chitosan (Q); AA, BB and CC: quadratic regression coefficients for G–G, W–W and C–C; AB, AC and BC: regression coefficients for interaction between G–W, G-Q and W-Q. * non-significant value (*p* < 0.05).

**Table 4 molecules-27-00869-t004:** Predicted and experimental responses of gelatin–whey protein–chitosan film prepared using the optimal FFS formulation.

Responses *	Predicted Value ^a^	Experimental Value (n = 3) ^b^	Absolute Residual Error (%) ^c^
MC (%)	18.99	19.12 ± 0.23	0.68
S (%)	26.81	25.25 ± 0.48	6.17
TR (%)	7.86	8.21 ± 0.06	11.57
YM	0.079	0.065 ± 0.017	21.54

* MC, moisture content; S, solubility; TR, transparency; YM, Young’s modulus. ^a^ Predicted values obtained from the model equations. ^b^ Experimental values obtained at optimum conditions (gelatin 6.0 g, whey 2.5 g and chitosan 1.5 g). ^c^ Absolute residual error (%) = [(experimental value − predicted value)/experimental value] × 100 [60]. R^2^ predicted and experimental values correlation (0.9982).

## Data Availability

Not applicable.

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
