# Peer review of "Optimization of the Physical, Optical and Mechanical Properties of Composite Edible Films of Gelatin, Whey Protein and Chitosan"

_molecules, 2022, doi:10.3390/molecules27030869_

Round 1
Reviewer 1 Report
Dear Authors,
Overall, this manuscript is interesting and covers an interesting topic. The concept of optimizing the production of edible membranes also seems to be right. However, this manuscript needs to be corrected
Detailed comments below:
Line 48: I think you should mention a few examples of products that use by-products from agri-food processing. You can show various possibilities of managing these raw materials (food additives, packaging, biocomposites, etc.). Then, in the next sentence, add that one of the ways of using these raw materials can be, for example, edible coatings ...... ..
Browse the following articles: "Development of polyvinyl alcohol and apple pomace bio-composite film with antioxidant properties for active food packaging application", "Properties of Biocomposites from Rapeseed Meal, Fruit Pomace and Microcrystalline Cellulose Made by Press Pressing: Mechanical and Physicochemical Characteristics", “Characterization of corn extrudates with the addition of brewers' spent grain as a raw material for the production of functional batters”, and others.
Line 85: Don't write here that compatibility assessment can be achieved by the SRM method. There are, after all, other satistic methods. You don't really know if this method is the best for describing these phenomena. Better not to highlight it.
Line 91: According to table 1 and the description of the above table should be included in the research methodology.
Line338: Are your results close to or far from expected values? Give some examples of research results, e.g. for commercial products. Review all test results for this.
Line 362: This section should be in place of the Test Results section, Line 90.
Line 390: Each scientific apparatus should be described as follows (model: manufacturer, city, country). Review the entire methodology.
Line 460: If possible, also refer to the research standard for determining MY. I think you should describe in more detail how this indicator is determined. Why not present the formulas used in the calculations? Based on this information, it is difficult to reproduce the results of the research.
Line 495: Conclusions need to be improved. You must add conclusions to the future. For example, write about the impact of your research on the further development of edible coatings ....
Author Response
January 21th, 2021
Response to reviewer 1
Molecules-1537613
Title: “Optimization of the physic, optical and mechanical properties of a composite edible films of gelatin, whey protein and chitosan”
This document contains our reply to comments from editor and reviewer 1 on Manuscript ID: molecules-1537613. We are sure that these changes as well as those made in answer to reviewer’ comments have greatly improved our manuscript, and would like to thank both the editor and reviewer for their kind help and suggestions.
Reviewer 1
Overall, this manuscript is interesting and covers an interesting topic. The concept of optimizing the production of edible membranes also seems to be right. However, this manuscript needs to be corrected
Detailed comments below:
Line 48: I think you should mention a few examples of products that use by-products from agri-food processing. You can show various possibilities of managing these raw materials (food additives, packaging, biocomposites, etc.). Then, in the next sentence, add that one of the ways of using these raw materials can be, for example, edible coatings ...
Browse the following articles: "Development of polyvinyl alcohol and apple pomace bio-composite film with antioxidant properties for active food packaging application", "Properties of Biocomposites from Rapeseed Meal, Fruit Pomace and Microcrystalline Cellulose Made by Press Pressing: Mechanical and Physicochemical Characteristics", “Characterization of corn extrudates with the addition of brewers' spent grain as a raw material for the production of functional batters”, and others
Reply: Thank you very much for the comment, we agree with you with, we considered the references that you suggested.
The most widely used materials for packaging food, cosmetics and pharmaceutical products are synthetic plastics [1,2], due to their low production cost, light weight, and excellent protection that they provide to the packaged product [3]. However, as they are synthetic compounds, with delayed biodegradation (100-700 years) and oil derivatives, their use poses serious ecological problems such as the emissions released during their manufacture and incineration, the use of resources and the generation of waste [4], therefore, there is an urgent need driven by both industry and consumers to develop environmentally friendly, renewable, naturally-derived polymeric materials involving processes that are profitable, minimize pollution and environmental emissions [5,6]. This type of natural polymers includes polysaccharides, proteins and lipids that have become an alternative eco-friendly material for potential elaboration in various applications, such as packaging, paper coating, biomedical, food science and agriculture [2], as well as edible films [7]. This type of raw materials includes various by-products of earlier technological processes related to agricultural and food processing [8]. Among the materials obtained from agroindustrial by-products, which are employed in the elaboration of edible packaging, extruded corn, post-extraction rapeseed meal, dried fruit pomace (chokeberry, currant, apple and raspberry pomace) and microcrystalline cellulose (MCC), whey protein, gelatin, chitosan, soy protein, zein, can be found, among others [8-14].
Line 85: Don't write here that compatibility assessment can be achieved by the SRM method. There are, after all, other satistic methods. You don't really know if this method is the best for describing these phenomena. Better not to highlight it.
Reply: We are very grateful for this observation, which gives a better meaning to the sentence and the meaning that it wants to give, we made the changes in the text:
Nonetheless, until this moment, there are no records on composite films consisting of gelatin, whey protein and chitosan, and thus, there is a need to study the relation between these components, since the composition of the film affects its structure and properties [27]; The SRM allows understanding the individual relationship and the interaction between multifactor based on physical responses. [43-45].
Line 91: According to table 1 and the description of the above table should be included in the research methodology.
Reply: Table 1 is already cited in methodology 3.13 experimental design, we made the changes in the text:
Each factor was tested at three levels low (-1), center (0) and high (1) (Table 1), the range of each component was chosen based on previous studies [11, 74] and preliminary trials.
Line 338: Are your results close to or far from expected values? Give some examples of research results, e.g. for commercial products. Review all test results for this.
Reply: The results of the optimization were close to the experimental in a 99.82%, which indicates that the optimization through this design can predict the behaviour of the films according to the composition, we made the changes in the text:
With the aim of finding the optimal formulation of the film, to achieve the desired value of the response variables, the simultaneous optimization of multiple responses function was employed. The optimization was carried out on the basis of the following objectives: 1) minimize the moisture content, 2) minimize solubility, 3) maximize transparency and 4) minimize Young’s modulus, intended to be used for high-moisture foods, however, this optimization can fit the desired goal, as long as the prediction is within experimentally observed parameters. The optimal level of the different parameters, via the application of the methodology of the desired function, in terms of real values of the raw materials, 6 g of gelatin, 2.5 g of whey protein and 1.48 g of chitosan, were obtained, with an estimated desired of 0.9258, subsequently, the estimated parameters were validated with new formulations under the conditions suggested above.
The validation of the model was carried out via the comparison of the predicted results with the experimental results of the edible film composed of C-G-W in optimized conditions (Table 4). The absolute residual error of the dependent variables was found to be between 0.68% and 21.54%, which indicates the validity of the responses obtained through the Box-Behnken experimental design and are found to be in conformity with the precision of models generated for other films [60-62], the correlation value between the predicted values and those obtained experimentally was 0.9982, which adequately validates the optimization model.
Similar results were found by [63], using glycerol, chitosan and drying temperature, were found out a desirability of 1.0, mean optimized values were for thickness, moisture, solubility, L*, a*, b*, penetrability, density, transmittance and WVTR. [60], using Box–Behnken response surface design got to maximize transparency and to minimize solubility, yellowness index and moisture content in a pea starch/guar gum edible film, with an overall desirability of 0.77, these characteristics can be effectively produced and successfully applied in the food packaging industry. On the other hand, [43], optimized a film formation following this consideration: (1) minimization of WVP; (2) maximization of tensile strength and (3) minimization of solubility in a sesame protein edible film, showing a good mechanical property and can be used for the packaging or coating applications of fruits and vegetables and provides partial barrier to moisture and considerate in reducing moisture loss. [44] observed that edible coatings based on chitosan, glycerol and carotenoproteins, the mechanical and the biological properties of composite films using RSM could be an attractive natural alternative against fungi that attack fruits and vegetables, thereby, preventing the occurrence of health and environmental problems. Other studies [64] achieved an optimization of chitosan, starch and glycerol for physical, mechanical and barrier properties by RSM showing that values for physical and mechanical properties were found to be similar to the predicted values, these results shows that this formulation can be applied to prepare the pea starch film with good physical and mechanical properties for further utilization.
As can be seen, the use of RSM for the elaboration of edible coating could predict future formulations and behaviors to provide alternatives for its application.
Line 362: This section should be in place of the Test Results section, Line 90.
Reply: We understand your concern about the order that the manuscript should have, and we would gladly do it instead, however, according to the journal's author guide, it is requested that the methodology come after the results.
Line 390: Each scientific apparatus should be described as follows (model: manufacturer, city, country). Review the entire methodology.
Reply: Thank ypu for the observations, we made the changes in the text:
…using a Brookfield Viscometer (Model RVDV-I, Brookfield Engineering Labs Inc., Mid-dleboro, MA, USA) with…
… chamber (ITH-75: Lumistell, Gto, México) at…
… a digital micrometer (RS Pro, Fujian, China)…
… a digital micrometer (RS Pro, Fujian, China)…
… spectrophotometer (UV-5500PC: M&A Instruments Inc, CA, USA)…
…, a texturometer (TA.XT Plus Texture Analyser: Stable Micro Systems, UK)…
Line 460: If possible, also refer to the research standard for determining MY. I think you should describe in more detail how this indicator is determined. Why not present the formulas used in the calculations? Based on this information, it is difficult to reproduce the results of the research.
Reply: All parameters were determined through the Exponent software version 6.12, to determine the maximum force at break, maximum distance at break, slope value / distance 6.25 mm up to 50% of the penetration distance of the punch, we made the changes in the text:
The mechanical properties (strength (ST) and Young’s modulus (MY)) were evaluated via the puncture test, according to the methodology described by [26], with some modifications (Figure 4). Briefly, a Stable Micro Systems TA-XT2 texturometer (Texture Technologies Corp.), with a P.0.255 ¼” stainless-steel spherical test probe, a test speed of 1.0 mm/s and a penetration distance of 30.0 mm, was employed to obtain the force-deformation curve (Figure 5). From these curves, the strength and Young’s modulus were calculated. The strength was considered the maximum force prior to the fracturing of the CEF. Young’s modulus (YM) was considered the slope of the linear proportion (6.25 mm up to 50% of the penetration distance of the probe) of the force-deformation curves. The measurements were repeated seven times for each composite edible film. All parameters were determined through the Exponent 6.1.20.0 (Stable Micro Systems, UK).
Figure 4. Schematic diagram of the texture analyzer used to compression test for film. The texture analyzer (a) consisted of a probe and a support. (b)System for measuring the elongation of composite films. D: Maximum penetration distance of the probe before rupture (mm). R: Radius of circumference of the plate (12.7 mm). E: Elongation (mm) of the film [73].
Figure 5. Force-deformation curve [74]
Line 495: Conclusions need to be improved. You must add conclusions to the future. For example, write about the impact of your research on the further development of edible coatings ....
Reply:
The design of the three-level Box-Behnken surface response methodology allowed the successful optimization of the optimal formulation of edible films composed of G-W-C. The concentration of three biopolymers significantly affected the viscosity of the film-forming solution and the physical, optical, and mechanical properties of the films. It was found that the optimal formulation of the films contained 6 g of gelatin, 2.5 g of whey protein and 1.48 g of chitosan, since they presented a lower moisture content, a greater solubility, greater transparency, and better mechanical properties than the rest of the formulations, which is suggested for high moisture foods. The present study suggests that films composed of G-W-C exhibit satisfactory properties for their use as biodegradable packaging materials. These films, being made from biodegradable, non-toxic, edible natural sources, can replace synthetic polymers in certain circumstances, which gives the opportunity to reduce the use of non-degradable synthetic materials, in addition to providing options for the development of new plastic products for the pharmaceutical, food and cosmetic industries, the use of RSM for the elaboration of edible coating could predict future formulations and behaviors to provide alternatives for its application. Nonetheless, additional studies are required to determine the use of the films in commercial food systems.
Thank you so much. We appreciate your time.
Best Regards
Octavio Dublán-García
Laboratorio de Alimentos y Toxicología Ambiental,
Facultad de Química
Universidad Autónoma del Estado de México

Reviewer 2 Report
Comments on molecules-1537613-v1
This paper is well written, however, its organisation should be optimized for making its understaning clearer. The subject of the study and the experimental approach are not at all original, the techniques are basic and not necessarily well mastered, or at least their interpretation should take into account many possible biases. The interpretations should therefore be reviewed in depth (see detailed comments below).
Therefore, I have reservations about the publication of this article and recommend "a major revision".
The Materials and Methods section should be placed before the results.
Introduction
L50 : whey protein, gelatin, chitosan, soy protein, zein etc are no more considered as waste, but as by-products. Nowadays, whey proteins as powder is no more considered as waste or by products, but as functional ingredient. Most of the cited are considered as food ingredients and not waste, which the cost/price is for instance from $20 to $50/kg for whey protein, i.e twice more expensive than milk powder ! Even gelatine is much more expensive than polyolefins. Therefore, the argument related to waste is wrong and is uncompetitive compared to plastics. I think the authors should more consider the environmental aspect of the end of life and food waste reduction by using edible films/coatings.
L74 : the main drawback of chitosan is its uncompetitive cost.
L76 : the statement is wrong, there is no « sole component » films in the literature, all contains plasticizers, residual water, sometimes crosslinkers etc.. The authors should say sole or mono biopolymer component. When several biopolymer mixed, they can be considered as composites, even if there is homogeneous (monophasic) system or IPN (interpenetrated network).
Materials and Methods Section
L365 : give purity of macromolecules : gelatin, whey protein and chitosan, the average MW, what is the gelatin origin (pork, fish, bovine, skin or bones or hydes..), from whey proetin, is whey obtain from acid curd process or from rennet cheese process. What is the nature of the 15% lactic acid impurities ?
Table 1 L97 : how the concentration range were chosen ? why content range for gelatin, chitosan and whey selected different according the biopolymer ? What are the justification. Is it related to the number of positive and negative charges in each biopolymer at pH 3.5 to enehance electrostatic interactions ?.
L417 water solubility. The pretreatment at 105°C for 24h may induced a Maillard process, according the presece of impurities in whey (lactose) and in lactic acid that may favorise Maillard reaction and thus crosslinking in proteins. This probably modified the solubility of unheated films. So solubility measured does not correspond to the neat films !
L426 : Was the partial pressure considered 0 % RH above silicagel ? Silicagel when perfectly and freshly dried, has a 0.5% Rh and very often when used around 1.5-2% Rh. This should be considered in the pressure differential. Was the thermodessiccator ventilated to prevent the stagnant layer ? If not, WVP values should be corrected according to the Mc Hugh (Hydrophilic Edible Films: Modified Procedure for Water Vapor Permeability and Explanation of Thickness Effects T. HABIG McHUGH,R. AVENA-BUSTILLOS,J.M. KROCHTA,J Food Sciences, July 1993 https://doi.org/10.1111/j.1365-2621.1993.tb09387.x)
L460 mechanical test : why tensile test not used as 99.9% of packaging films are characterized and according all standards of mechanical chracaterization of packaging films ? TAXT2 allows that measurements. When penetration test were done, if diameter of film is 5 cm according the etri dishes used, and a probe of ¼", how was fixed the films on support ? What was the shape of the support (cylinder, empty cube )? Was the border effect considred as the distance between probe and border is very small ?
Results and discussions.
Table 1 should be cited in the material and methods section, when model and concentration ranges are cited or must be cited and justified
In table 2, please give the mean+Stad deviation of responses, or indicates with stars the values significantly different at 5% and 1%. This table is just cited in the discussion, but not discussed. At least the range of response values should be discussed and compared to literature.
In table 3, please correct WAP by WVP
L168 viscosity is usually expressed in mPa.s instead of Cps, taht is the old unit system for viscosity.
L176-178 « described that the G-Q interactions are due to electrostatic attractive forces, covalent bonds, hydrogen bonds, dipole forces, among others » => please go further in discussion of interaction. Refers to pI of proteins and pKa of chitosan according the used pH to intrepret. Just citing the possible low energy bonds and interaction (almost all existing) did not provied interetsing knowledge. This kind of assumption is obvious and apply to any food complex system.
L178 «… being an ampholytic protein, can form a coacervated complex with the chitosan » But this depends on DA of chitosan, and thus of its pKa and of the pH, same thinking with proteins and pI.
L196 WVP : in the range of 10-50µm, the relation between WVTR and thickness is not lear and thus WVP of film in that range could only be discussed of compared with composition changed when thickness is very similar (see different paper of the thickness effect on WVP of polysacchraide or protein based films, at least 25 papers since 1993). Therefore, I suspect the effect of film composition discussed here is more related to the thickness variation and really to the composition.
L217: also for WAC, discussion refers to interaction, but remains very superficial, please be more accurate in nature and strength of interactions/bonds involved; Never forget also the possible role of impurities, probably very high influence if salts or reducing sugars were present, even at traces levels.
L235: “the increase in the concentration of chitosan promoted the formation of a compact and strong network, due to the interaction of the hydrogen bonds between the whey protein and chitosan”. What proof that hydrogen ponds is mainly responsible of such behavior ? whay not electrostatic, ionic or hydrophobic bonds less involved ? Why not assumptions of various IPN 5inter-penetrated Network) not suggested and discussed; hat is also possible (see literature in IPN made from whey protein and other biopolymers).
L256-302 Optical properties : there is no discussion on the isotropy or anisotropy of the network, crystalline fractions, IPN structure etc.. as mains factor influencing the transparency and light transmission/color of biopolymer films. See literature on film of coating not for food packaging or edible coating, but from varnishes and paints, or windows or glass lenses. This may helps to interpretation of optic properties.
L328-330 : discussion refres to irregularity or surface heterogeneity from literature to explain. Was surface characterized by microscopic observations, or by surface measurements?
L332-337 I don’t agree with the interpretation of plasticization process caused by chitosan. Plasticization is confirmed when Strength or YM decreases whereas elongation/deformation increases, but here we have no information on elongation/deformation. Usually, when plasticization occurs, both strength and YM evolves in same trends, that is not the case. In my mind the beahviour is more linked to a lubrication process, probably of IPN, in which the meshes of the IPN are different in size and type of stabilization interactions, and thus both network (chitosan on one side, and protein in the other side) moved partly independently from each over.
L340 : the optimization and validation of the design should be discussed at first, before discussion of physical-chemical parameter responses
L496-505 Conclusions should considered the previous comments and thus be more moderated.
Author Response
January 21th, 2021
Response to reviewer 2
Molecules-1537613
Title: “Optimization of the physic, optical and mechanical properties of a composite edible films of gelatin, whey protein and chitosan”
This document contains our reply to comments from editor and reviewer 1 on Manuscript ID: molecules-1537613. We are sure that these changes as well as those made in answer to reviewer’ comments have greatly improved our manuscript, and would like to thank both the editor and reviewer for their kind help and suggestions.
Reviewer 2
This paper is well written, however, its organisation should be optimized for making its understaning clearer. The subject of the study and the experimental approach are not at all original, the techniques are basic and not necessarily well mastered, or at least their interpretation should take into account many possible biases. The interpretations should therefore be reviewed in depth (see detailed comments below).
Therefore, I have reservations about the publication of this article and recommend "a major revision".
The Materials and Methods section should be placed before the results.
Reply: We understand your concern about the order that the manuscript should have, and we would gladly do it instead, however, according to the journal's author guide, it is requested that the methodology come after the results.
L50 : whey protein, gelatin, chitosan, soy protein, zein etc are no more considered as waste, but as by-products. Nowadays, whey proteins as powder is no more considered as waste or by products, but as functional ingredient. Most of the cited are considered as food ingredients and not waste, which the cost/price is for instance from $20 to $50/kg for whey protein, i.e twice more expensive than milk powder ! Even gelatine is much more expensive than polyolefins. Therefore, the argument related to waste is wrong and is uncompetitive compared to plastics. I think the authors should more consider the environmental aspect of the end of life and food waste reduction by using edible films/coatings.
Replay: Thank you very much for the comment, we agree with you with the use of words, as you mentioned, they are no longer considered as waste but as by-products, we changed the word and added the explanation:
The most widely used materials for packaging food, cosmetics and pharmaceutical products are synthetic plastics [1,2], due to their low production cost, light weight, and excellent protection that they provide to the packaged product [3]. However, as they are synthetic compounds, with delayed biodegradation (100-700 years) and oil derivatives, their use poses serious ecological problems such as the emissions released during their manufacture and incineration, the use of resources and the generation of waste [4], therefore, there is an urgent need driven by both industry and consumers to develop environmentally friendly, renewable, naturally-derived polymeric materials involving processes that are profitable, minimize pollution and environmental emissions [5,6]. This type of natural polymers includes polysaccharides, proteins and lipids that have become an alternative eco-friendly material for potential elaboration in various applications, such as packaging, paper coating, biomedical, food science and agriculture [2], as well as edible films [7]. This type of raw materials includes various by-products of earlier technological processes related to agricultural and food processing [8]. Among the materials obtained from agroindustrial by-products, which are employed in the elaboration of edible packaging, extruded corn, post-extraction rapeseed meal, dried fruit pomace (chokeberry, currant, apple and raspberry pomace) and microcrystalline cellulose (MCC), whey protein, gelatin, chitosan, soy protein, zein, can be found, among others [8-14].
L74 : the main drawback of chitosan is its uncompetitive cost.
Reply: Thank you for your comment, although the costs of chitosan are considered high, is a non-toxic polysaccharide, which is obtained from the deacetylation of the chitin that is extracted from the exoskeleton of crustaceans, insects, and the cell wall of microorganisms [20, 28]. It is a co-polymer of N-acetyl-D-glucosamine and D-glucosamine, with the latter being superior to 60% [29,30]. The biodegradability, biocompatibility, excellent film formation capacity [13,14] and its antimicrobial properties [32], which could explain its use in a wide range of applications that may include areas such as pharmaceuticals, biotechnology, biomaterials, biomedicine, agriculture and food processing [33], so using it as a material for the development of a biodegradable bioplastic is a cost benefit with a great impact on the environment compared to polyethylene-based plastics that take up to 700 years to degrade [4]. The biodegradable plastics market is projected to grow from over 2.0 billion USD in 2015 to 3.4 billion USD by 2020 with a growth rate of 10.8% per year [34,35].
L76 : the statement is wrong, there is no « sole component » films in the literature, all contains plasticizers, residual water, sometimes crosslinkers etc.. The authors should say sole or mono biopolymer component. When several biopolymer mixed, they can be considered as composites, even if there is homogeneous (monophasic) system or IPN (interpenetrated network).
Reply: We greatly appreciate this observation, which gives a better sense of what is meant, we make the correction in the text:
Coatings with 2 independent components have been studied [36,37] but until now the interaction of multicomponents in a film has not been studied.
Materials and Methods Section
L365 : give purity of macromolecules : gelatin, whey protein and chitosan, the average MW, what is the gelatin origin (pork, fish, bovine, skin or bones or hydes..), from whey proetin, is whey obtain from acid curd process or from rennet cheese process. What is the nature of the 15% lactic acid impurities?
Reply: We appreciate this observation, the characteristics of each macromolecule used were placed in the text:
The origin of the gelatin was placed; the method of obtaining serum was established; placed the impurities that lactic acid presents.
The materials used were bovine gelatin (Duche; 290° Bloom; México), whey protein (from the rennet cheese process) (Darigold Inc.; Seattle, WA, USA) and commercial-grade chitosan (Aldrich, Germany; low molecular weight; molecular mass of 50,000-190,000 Da; degree of deacetylation > 75%). All the chemical compounds and solvents used were of analytical grade and were obtained from Merck. Glycerol (J.T. Baker, Mexico) and 85% lactic acid (85-90% lactic acid, ACS - Chloride (Cl) ≤ 0.001 %, Trace Impurities - ACS - Heavy Metals (as Pb) ≤ 5 ppm, ACS - Sulfate (SO₄) ≤ 0.002 %, ACS - Chloride (Cl) ≤ 0.001 %) The reagent generally available is a mixture of lactic acid (CH₃CHOHCOOH) and lactic acid lactate (C₆H₁₀O₅) (J.T. Baker, Mexico) were employed to improve the mechanical properties of the film and increase the solubility of the polymer powders.
Table 1 L97: how the concentration range were chosen? why content range for gelatin, chitosan and whey selected different according the biopolymer ? What are the justification. Is it related to the number of positive and negative charges in each biopolymer at pH 3.5 to enehance electrostatic interactions ?.
Reply: Thank you very much for this comment, the interval of each component was chosen based on previous studies [11, 74], as well as a simplex centroid design, also, the low level was limited by the mechanical properties of the final composite film (films prepared from film-forming solutions containing a low amount of raw materials, which tend to be mechanically unstable and brittle), while the high level was limited by the physical properties of the films, i.e. the addition of raw materials in high concentrations give very viscous solutions that are difficult to apply.
pH 3.5 was used in all tests to avoid pH effect and complex coacervation and precipitation in case of an opposite charge of two polymers, and to improve chitosan solubility.
So the final paragraph was as follows:
In 100 mL of distilled water, at 70°C with constant magnetic stirring, chitosan was added and was taken to a pH of 3.5 with lactic acid, work was done in all the tests to avoid pH effects and complex coacervation and precipitation in case of an opposite charge of two polymers [40], and to improve the solubility of chitosan. Afterwards, gelatin, whey protein and glycerol were added, one by one, allowing the proper incorporation of each of the components (approximately 5-7 min between each one). Temperature and pH were always kept constant. The solution was cooled until it reached a temperature of 40°C, under constant stirring, and was identified as film-forming solution (FFS). The concentrations of each of the components are described in Table 1.
L417 water solubility. The pretreatment at 105°C for 24h may induced a Maillard process, according the presece of impurities in whey (lactose) and in lactic acid that may favorise Maillard reaction and thus crosslinking in proteins. This probably modified the solubility of unheated films. So solubility measured does not correspond to the neat films !
Reply: All the films were treated under the same conditions, following the methodology proposed in the literature [7,69], where they suggest working under these conditions (105°C, 24 hrs) for water solubility. We consider that as all the samples undergo the same conditions, the resulting solubility is dependent on the factors studied.
L426 : Was the partial pressure considered 0 % RH above silicagel ? Silicagel when perfectly and freshly dried, has a 0.5% Rh and very often when used around 1.5-2% Rh. This should be considered in the pressure differential. Was the thermodessiccator ventilated to prevent the stagnant layer ? If not, WVP values should be corrected according to the Mc Hugh (Hydrophilic Edible Films: Modified Procedure for Water Vapor Permeability and Explanation of Thickness Effects T. HABIG McHUGH,R. AVENA-BUSTILLOS,J.M. KROCHTA,J Food Sciences, July 1993 https://doi.org/10.1111/j.1365-2621.1993.tb09387.x)
Reply: Equation 1, shown, is the result of the composition of the 3 equations indicated in the literature (ASTM and T. HABIG McHUGH, R. AVENA-BUSTILLOS, JM KROCHTA, J Food Sciences, July 1993 https://doi.org/ 10.1111/j.1365-2621.1993.tb09387.x), so the conditions you mention were considered.
L460 mechanical test: why tensile test not used as 99.9% of packaging films are characterized and according all standards of mechanical chracaterization of packaging films ? TAXT2 allows that measurements. When penetration test were done, if diameter of film is 5 cm according the etri dishes used, and a probe of ¼", how was fixed the films on support ? What was the shape of the support (cylinder, empty cube )? Was the border effect considred as the distance between probe and border is very small ?
Reply: Based on how the films were made (circular shape), an adaptation of the method for determining the mechanical parameters was made as shown in figures 4 and 5, and changes were made to the text as shown below:
The mechanical properties (strength (ST) and Young’s modulus (MY)) were evaluated via the puncture test, according to the methodology described by [26], with some modifications (Figure 4). Briefly, a Stable Micro Systems TA-XT2 texturometer (Texture Technologies Corp.), with a P.0.255 ¼” stainless-steel spherical test probe, a test speed of 1.0 mm/s and a penetration distance of 30.0 mm, was employed to obtain the force-deformation curve (Figure 5). From these curves, the strength and Young’s modulus were calculated. The strength was considered the maximum force prior to the fracturing of the CEF. Young’s modulus (YM) was considered the slope of the linear proportion (6.25 mm up to 50% of the penetration distance of the probe) of the force-deformation curves. The measurements were repeated seven times for each composite edible film. All parameters were determined through the Exponent 6.1.20.0 (Stable Micro Systems, UK).
Figure 4. Schematic diagram of the texture analyzer used to compression test for film. The texture analyzer (a) consisted of a probe and a support. (b)System for measuring the elongation of composite films. D: Maximum penetration distance of the probe before rupture (mm). R: Radius of circumference of the plate (12.7 mm). E: Elongation (mm) of the film [73].
Figure 5. Force-deformation curve [74]
Results and discussions.
Table 1 should be cited in the material and methods section, when model and concentration ranges are cited or must be cited and justified
Reply: Thank you for the observation: Table 1 is cited in the methodology, in section 3.2 3.2. Preparation of the film-forming solution and 3.13 Experimental design.
In table 2, please give the mean+Stad deviation of responses, or indicates with stars the values significantly different at 5% and 1%. This table is just cited in the discussion, but not discussed. At least the range of response values should be discussed and compared to literature.
Reply: Standard deviations were added to the response table; however, they were not placed at the beginning to consider the size of the tableIn addition, the table corresponds to the descriptive summary of each response, the MRS methodology does not look for significant differences between the observed samples, its objective is to estimate regression models through said observations.
In table 3, please correct WAP by WVP
Reply: Correction was made in Table 3.
L168 viscosity is usually expressed in mPa.s instead of Cps, taht is the old unit system for viscosity.
Reply: Thank you very much for the comment, the correction was made in the document.
L176-178 « described that the G-Q interactions are due to electrostatic attractive forces, covalent bonds, hydrogen bonds, dipole forces, among others » => please go further in discussion of interaction. Refers to pI of proteins and pKa of chitosan according the used pH to intrepret. Just citing the possible low energy bonds and interaction (almost all existing) did not provied interetsing knowledge. This kind of assumption is obvious and apply to any food complex system.
L178 «… being an ampholytic protein, can form a coacervated complex with the chitosan » But this depends on DA of chitosan, and thus of its pKa and of the pH, same thinking with proteins and pI.
Reply: Thank you very much for the comment, for a better explanation, the following was added:
The viscosity of the film-forming solutions was found within an interval of 26.53 to 542.53 mPa.s. The mathematical model showed that the linear, interaction and quadratic effects were significant about the viscosity results, except for the quadratic effect of gelatin and the W-C interaction, which did not show a significant effect (P < 0.05). Factors G, W and C and interactions G-W, G-C, G-G and W-W showed a positive effect over viscosity (Figure 1A), where the C-W interactions is observed, [15] have reported that the concentration of protein in film-forming solutions affects viscosity, due to the interactions between the polymers, while the addition of chitosan to the solution interferes with the formation of the protein network, because of the G-C interactions. the enhanced mechanical properties of protein– polysaccharide blends are attributed to the multiple and strong intermolecular interactions (by hydrogen bonding, dipole– dipole link formation, and charge effects) between hydroxyl groups of the polymer chains. In addition, cross-linking with thermal treatment allows the generation of bonds between the chains of proteins and polysaccharides (products of Maillard reactions) which improve the mechanical properties of the polymer network [34,37], hydrophobic forces could play a major role in the binding between C and W according to [46,47], at pH values lower than 6.3, the amino groups are protonated ( NH3+), which allows C to make electrostatic contacts with anionic proteins, binding to the OH- groups of the proteins [48], which leads to an increase in the viscosity of the composite solution.
L196 WVP : in the range of 10-50µm, the relation between WVTR and thickness is not lear and thus WVP of film in that range could only be discussed of compared with composition changed when thickness is very similar (see different paper of the thickness effect on WVP of polysacchraide or protein based films, at least 25 papers since 1993). Therefore, I suspect the effect of film composition discussed here is more related to the thickness variation and really to the composition.
Reply: The values were calculated considering as a correction factor, the thickness of each sample in order to eliminate the effect of the differences between the thicknesses of each EC, for this reason, the correction was carried out using the adequacy made by T. HABIG McHUGH, to the methodology of the ASTM [70].
L217: also for WAC, discussion refers to interaction, but remains very superficial, please be more accurate in nature and strength of interactions/bonds involved; Never forget also the possible role of impurities, probably very high influence if salts or reducing sugars were present, even at traces levels.
Reply: In order to complement the explanation of the interaction, the following was added:
Chitosan exhibits poor solubility in neutral and basic media, limiting its use in such conditions, but is soluble in aqueous acidic media via primary amine protonation [50], the presence of large amounts of protonated -NH2 groups on the chitosan structure accounts for its solubility in acid aqueous media since its pKa value is approximately 6.5 [51]
The possible interaction with impurities was placed within this explanation. At the moment there is no impurity analysis of the raw materials, since they were obtained from commercial suppliers of chemical products and reagents such as Aldrich, J.T. Baker, but we will consider performing this analysis in future research to be sure that these trace compounds could intervene in the result.
L235: “the increase in the concentration of chitosan promoted the formation of a compact and strong network, due to the interaction of the hydrogen bonds between the whey protein and chitosan”. What proof that hydrogen ponds is mainly responsible of such behavior ? whay not electrostatic, ionic or hydrophobic bonds less involved ? Why not assumptions of various IPN (inter-penetrated Network) not suggested and discussed; hat is also possible (see literature in IPN made from whey protein and other biopolymers).
Reply: Thank you very much for the comment, for a better explanation, the following was added:
this could be due to different behaviors, such as the formation of an interpenetrating polymer network, which is a combination of two crosslinked polymers [52], films con-sisting of two or more polymer networks interpenetrate providing a first brittle network, comprising densely crosslinked polyelectrolyte chains, while the second is ductile and comprises weakly crosslinked nonionic polymer chains, which allows outstanding me-chanical properties to be obtained that are much better than the sum of the mechanical properties of its network components separately [53], for example, in previous studies carried out in an experiment with mixtures, it was observed that 100% chitosan films were weaker than those with the mixture of 50% protein and 50% chitosan; or
L256-302 Optical properties: there is no discussion on the isotropy or anisotropy of the network, crystalline fractions, IPN structure etc.. as mains factor influencing the transparency and light transmission/color of biopolymer films. See literature on film of coating not for food packaging or edible coating, but from varnishes and paints, or windows or glass lenses. This may help to interpretation of optic properties.
Reply: Thank you very much for the comment, to complement the explanation, the crystallinity part of the samples was added:
The crystallinity of the samples could result from the interaction between chains through intermolecular hydrogen bonds and van der Waals bonds that form between the surface of a compound and C (cationic behavior), since it is a mechanical type of bond. which occurs due to the charges of both materials (electrostatic adhesion), [58], observed this behavior when hydroxyapatite and C.
L328-330 : discussion refres to irregularity or surface heterogeneity from literature to explain. Was surface characterized by microscopic observations, or by surface measurements?
Reply: Thank you very much for the comment, the results of that study show the microstructure, but the purpose of citing this work is that, by having a chitosan-whey protein interaction, they obtained weaker films, so the sentence was rewritten as follows:
while the presence of whey protein film mixed with chitosan obtained weaker films [40]
L332-337 I don’t agree with the interpretation of plasticization process caused by chitosan. Plasticization is confirmed when Strength or YM decreases whereas elongation/deformation increases, but here we have no information on elongation/deformation. Usually, when plasticization occurs, both strength and YM evolves in same trends, that is not the case. In my mind the beahviour is more linked to a lubrication process, probably of IPN, in which the meshes of the IPN are different in size and type of stabilization interactions, and thus both network (chitosan on one side, and protein in the other side) moved partly independently from each over.
Reply: We completely agree with your observation, the correction was made in the wording, based on the results of table 2, since more elastic materials require less deformation stress, on the contrary, more resistant materials exhibit more elastic moments. little ones. So, the paragraph looks like this:
The gelatin and chitosan improved the Young’s modulus of the films, from 0.06 to 0.98. Both independent variables showed a positive effect over Young’s modulus, which means that increasing the content of chitosan and gelatin decreased the elasticity of the polymeric network, making the film more resistant to traction (Figure 3L), [5] showed an opposite effect, that addition of chitosan to the gelatin films produced more flexible films, which suggests that chitosan participates in the debilitation or reduction in the number of hydrogen bonds, acting as a plasticizer.
L340 : the optimization and validation of the design should be discussed at first, before discussion of physical-chemical parameter responses
Reply: We completely agree with your observation, design optimization and validation should be discussed first, before discussion of physico-chemical parameter responses, so the order goes as proposed in the document.
L496-505 Conclusions should considered the previous comments and thus be more moderated.
Reply: Thank you very much for the comment, the conclusions were changed:
The design of the three-level Box-Behnken surface response methodology allowed the successful optimization of the optimal formulation of edible films composed of G-W-C. The concentration of three biopolymers significantly affected the viscosity of the film-forming solution and the physical, optical, and mechanical properties of the films. It was found that the optimal formulation of the films contained 6 g of gelatin, 2.5 g of whey protein and 1.48 g of chitosan, since they presented a lower moisture content, a greater solubility, greater transparency, and better mechanical properties than the rest of the formulations, which is suggested for high moisture foods. The present study suggests that films composed of G-W-C exhibit satisfactory properties for their use as biodegradable packaging materials. These films, being made from biodegradable, non-toxic, edible natural sources, can replace synthetic polymers in certain circumstances, which gives the opportunity to reduce the use of non-degradable synthetic materials, in addition to providing options for the development of new plastic products for the pharmaceutical, food and cosmetic industries. Nonetheless, additional studies are required to determine the use of the films in commercial food systems.
Thank you so much. We appreciate your time.
Best Regards
Octavio Dublán-García
Laboratorio de Alimentos y Toxicología Ambiental,
Facultad de Química
Universidad Autónoma del Estado de México
Please see the attachment.

Round 2
Reviewer 1 Report
Dear Authors,
After taking into account the comments, the manuscript looks good, mainly from the scientific point of view. Congratulations on a good article.
Reviewer 2 Report
The authors well respected the comments and give argued answers; Therefore, The manuscript is now able to be accepted